# Transcriptomics Profiling of *Acer pseudosieboldianum* Molecular Mechanism against Freezing Stress

**DOI:** 10.3390/ijms232314676

**Published:** 2022-11-24

**Authors:** Zhiming Han, Xiangzhu Xu, Shikai Zhang, Qiushuang Zhao, Hanxi Li, Ying Cui, Xiao Li, Liran Wang, Su Chen, Xiyang Zhao

**Affiliations:** 1State Key Laboratory of Tree Genetics and Breeding, Northeast Forestry University, Harbin 150040, China; 2Jilin Provincial Key Laboratory of Tree and Grass Genetics and Breeding, College of Forestry and Grassland Science, Jilin Agricultural University, Changchun 130118, China; 3College of Veterinary Medicine, Jilin University, Changchun 130062, China

**Keywords:** *Acer pseudosieboldianum*, freezing stress, RNA-seq, DEGs

## Abstract

Low temperature is an important environmental factor that affects the growth and development of trees and leads to the introduction of failure in the genetic improvement of trees. *Acer pseudosieboldianum* is a tree species that is well-known for its bright red autumn leaf color. These trees are widely used in landscaping in northeast China. However, due to their poor cold resistance, introduced *A. pseudosieboldianum* trees suffer severe freezing injury in many introduced environments. To elucidate the physiological indicators and molecular mechanisms associated with freezing damage, we analyzed the physiological indicators and transcriptome of *A. pseudosieboldianum*, using kits and RNA-Seq technology. The mechanism of *A. pseudosieboldianum* in response to freezing stress is an important scientific question. In this study, we used the shoots of four-year-old *A. pseudosieboldianum* twig seedlings, and the physiological index and the transcriptome of *A. pseudosieboldianum* under low temperature stress were investigated. The results showed that more than 20,000 genes were detected in *A. pseudosieboldianum* under low temperature (4 °C) and freezing temperatures (−10 °C, −20 °C, −30 °C, and −40 °C). There were 2505, 6021, 5125, and 3191 differential genes (DEGs) between −10 °C, −20°C, −30°C, −40 °C, and CK (4 °C), respectively. Among these differential genes, 48 genes are involved in the MAPK pathway and 533 genes are involved in the glucose metabolism pathway. In addition, the important transcription factors (MYB, AP2/ERF, and WRKY) involved in freezing stress were activated under different degrees of freezing stress. A total of 10 sets of physiological indicators of *A. pseudosieboldianum* were examined, including the activities of five enzymes and the accumulation of five hormones. All of the physiological indicators except SOD and GSH-Px reached their maximum values at −30 °C. The enzyme activity of SOD was highest at −10 °C, and that of GSH-Px was highest at −20 °C. Our study is the first to provide a more comprehensive understanding of the differential genes (DEGs) involved in *A. pseudosieboldianum* under freezing stress at different temperatures at the transcriptome level. These results may help to clarify the molecular mechanism of cold tolerance of *A. pseudosieboldianum* and provide new insights and candidate genes for the genetic improvement of the freezing tolerance of *A. pseudosieboldianum*.

## 1. Introduction

Maple is a perennial deciduous tree or shrub of the genus maple in the family *Aceraceae*, with a natural distribution in Asia, Europe, North America, and the northern edge of Africa [1,2,3]. China is the main distribution area of maple species with rich genetic material resources, with more than 150 species of the genus maple, accounting for more than half of the world’s maple resources. This is significant for research into the evolutionary history of maple species [4,5]. In past years, these species of the genus maple have been widely introduced, domesticated, and utilized, due to their adaptability, resistance, ornamental value, and economic value [6,7,8]. Maple is widely used in the timber and pharmaceutical industries. It is a well-known modern landscape species, with large trees, beautiful foliage, colorful leaves, peculiar fruit shapes, and rich genetic variation, and it occupies an important position in urban ecosystems and landscape construction [9,10]. 

Through previous studies, extracts of *Acer* roots, leaves, bark, fruits, and seeds have been found to contain rich nutrients, such as amino acids, fatty acids, and mineral elements, as well as some key physiologically active substances, such as triterpenoids, chlorogenic acids, and neuronic acids [11]. In addition to their medicinal value, many *Acer* species have a rich variety of leaf color types during growth and development, such as from green to yellow (such as *A. catalpifolium* Rehde. and *A. truncatum* Bunge) and green to red (such as *A. mandshuricum* Maxim. and *A. pseudosieboldianum*). In addition, *Acer* species have a rich phenotypic variation in leaf shape, and three-, five-, and seven-leaved simple leaves can be found in *Acer* species, laying the foundation for further genetic improvement of *Acer* species [12,13,14]. 

However, most *Acer* trees are in danger of extinction due to frequent human activities and pests. Currently, most *Acer* tree species, such as *A. campestre L.*, *A. yangbiense* Y.S.Chen & Q.E.Yang, and *A. miaotaiense* P.C. Tsoong, are listed as endangered by the International Union for Conservation of Nature and Natural Resources (IUCN) Red List of Threatened Species, and these *Acer* species are also listed as national protected plants in China [15,16].The valuable deciduous tree species *A. pseudosieboldianum* (2n = 2x = 16) is only fragmented in northeastern China, the Russian Far East, and the northern half of the Korean Peninsula [17,18]. It is regarded as one of the most promising native trees for landscaping in northeastern China, due to its use as an essential leaf plant, its gorgeous winged fruit, and its intensely crimson leaves in autumn.

Hypothermia, as an abiotic stress, not only affects the geographical distribution of many important crops, but also negatively affects crop yield and quality every year [19,20]. Hypothermia stress is classified as cold (0–15 °C) and freezing (<0 °C) and has a significant impact on plant survival and geographic distribution [21]. A large number of plant species grow in tropical and subtropical regions, and most of them are sensitive to cold. However, plants can increase their frost tolerance through cold acclimation [22,23].

Previous studies have shown that freezing stress can trigger the activation of defense systems in plants. The peroxidase (POD), catalase (CAT), superoxide dismutase (SOD), and aseorbateperoxidase (APX) protective enzyme activity is enhanced, reducing the accumulation of reactive oxygen species (ROS) and reducing the .freezing stress damage to plants [24]. In plant endogenous hormones, abscisic acid (ABA) and proline act as important regulators of plant response to adversity stress, participating in this defense process. ABA is involved in plant growth and development regulation and responses to different environmental stresses, and freezing stress can promote the accumulation of ABA [25]. In terms of molecular regulation, the response of plants to freezing stress is mainly through the sensing and transduction of low temperature signals by cells to regulate gene expression, thereby regulating different metabolic pathways and signal transduction pathways, and responding at the transcriptional and translational levels [26].

There are thousands of genes that regulate these metabolites through upregulation and downregulation in freezing stress, and researchers have found that cold environments cause numerous alterations in the physiology and gene expression patterns that include carbohydrate metabolism, molecular chaperones, antifreeze proteins, signal transduction (receptor kinases, phosphatases, and Ca^2+^ binding proteins), and regulatory proteins [27,28,29,30]. The mechanisms of woody plant responses to freezing stress have drawn increased attention in recent years. By using transcriptomic and multi-omics analysis, it was possible to identify numerous cold-responsive genes that are involved in photosynthesis, calcium signaling, ABA homeostasis and transport, and antioxidant defense systems. For instance, the transcriptome alterations of *P. simonii* under cold stress were investigated. Microarray analysis identified many cold response genes involved in photosynthesis, Ca^2+^ signaling, ABA homeostasis and transport, and antioxidant defense systems [31]. The responses of *P. tremula* L. and *P. tremuloides Michaux*to cold stress have also been studied. Four different soluble carbohydrates (glucose, fructose, sucrose, and alginate) were found to accumulate rapidly under cold treatment, and proteomic analysis detected many important proteins in this process, such as chaperone proteins, dehydrins, and late-embryonic-development-enriched proteins [32,33]. 

In northeastern China, *A. pseudosieboldianum* is mainly found in the cold-temperate zone and can withstand temperatures below 30 °C in winter. It has unique adaptability and cold resistance compared with related Acer species [34,35]. *A. pseudosieboldianum* may be a potential resource of red-leaved maple trees for landscaping and future hybridization studies of the genus Macrocystis. Currently, the species is listed as endangered by the Chinese government and is protected ex situ, making *A. pseudosieboldianum* extremely valuable for research aimed at development and utilization.

Low temperature is an important environmental factor that affects the growth and development of trees and leads to the introduction of failure in the improvement of tree genetics. To elucidate the physiological indicators and molecular mechanisms associated with freezing damage, we analyzed the physiological indicators and transcriptome of *A. pseudosieboldianum*, using kits and RNA-Seq technology. In this experiment, four-year-old *A. pseudosieboldianum* twigs were used as samples, and the genes of *A. pseudosieboldianum* under freezing stress were studied through transcriptome.

## 2. Results

### 2.1. Measurement and Analysis of A. pseudosieboldianum Physiological Indicators

The production of reactive oxygen species is unavoidable in the normal life activities of plants, while reactive oxygen species scavenging systems exist in plants to maintain normal physiological activities. When plants are under adverse conditions, such as low temperature or drought, the balance between the intracellular production and the scavenging of reactive oxygen species is disrupted, and the increase of reactive oxygen species leads to cell injury. 

In this study, we measured the activities of SOD, POD, CAT, APX, and GSH-Px and measured ABA, MDA, H_2_O_2_, proline, and soluble sugar by the enzymatic activities of five enzymes and the degree of accumulation of five hormones and metabolites to observe how various physiological indicators accumulated under freezing stress at different temperatures (Figure 1). As shown in Figure 1, the activities or concentrations of the hormones reached their highest values under freezing stress at −30 °C, except for SOD and GSH-Px; SOD reached its highest activity under freezing stress at −10 °C, and GSH-Px reached its highest activity under freezing stress at −20 °C.

### 2.2. Illumina RNA-Seq and Align Analysis

The transcriptome sequencing analysis of 15 samples was completed and a total of 135.57 Gb clean data were obtained. The percentage of clean data of each sample reached 7 Gb, Q30 bases, and the GC content exceeded 92%. The GC content was more than 44%, and the genomic spectrum rate was more than 73%, reaching 87.19%, and 91.24%, respectively (Table 1 and Appendix A). The average unique mapped reads reached 84.57%, which is enough for subsequent bioinformatics analysis [27,32].

### 2.3. Identification and Analysis of DEGs

In this experiment, DESeq2 was used to analyze the differentially expressed genes, and the total number of differentially expressed genes, the number of upregulated genes, and the number of downregulated genes in each group were counted (Appendix A). By comparing the *A.pseudosieboldianum* of different periods, we studied the specific DEGs of freezing stress at different temperatures. At 4 °C, −10 °C, −20 °C, −30 °C, and −40 °C, and at 4°C as the CK group, 2505 (996 upregulated genes and 1509 downregulated genes), 6021 (2587 upregulated genes and 3434 downregulated genes), 5125 (2095 upregulated genes and 3030 downregulated genes), and 3191 (1068 upregulated genes and 2123 downregulated genes) DEGs were identified, respectively, in comparison with the CK group (Figure 2C).

As shown in Figure 3, 870 common DEGs were detected under five different degrees of freezing stress, which may indicate that these DEGs were involved in the treatment of low temperatures at different low temperatures, and all participated in the same pathway in response to freezing stress. In addition, at −30 °C, the number of DEGs only (2002) was significantly higher than that of other groups, indicating that these genes may be related to the difference in cold tolerance of *A. pseudosieboldianum*.

In this study, the total RNA from the branch was used for qRT-PCR verification. As shown in Figure 3, the six DEGs showed similar expression patterns between qRT-PCR data and RNA-seq results, proving that RNA-seq data are highly reliable for further analysis.

### 2.4. Notes and Enrichment of A. pseudosieboldianum Transcriptome

To understand the function of DEGs found between the two cultivars, all DEGs and protein databases were searched, and then gene ontology (GO) analysis was carried out to evaluate the function of genes. Gene Ontology [36] (GO) is the international standard classification system of gene function. As a database established by the Gene Onotology Consortium, it aims to establish a language vocabulary standard that is suitable for various species, defines and describes the function of genes and proteins, and can be updated with the deepening of research. GO is divided into three parts: molecular function, biological process, and cellular component. In the GO terms of each treatment, the greater the proportion of DEGs with stronger freezing stress, the more changes that were made in the metabolic process (Figure 4). In the comparative analysis of the five groups, the DEGs of these GO terms were divided into “metabolic process”, “cell process”, and “response to stimulation”; in the cellular component, they were divided into “cell”, “cell part”, and “membrane”; and in molecular function, they were divided into “catalytic activity”, “binding”, and “transporter activity”.

In addition, to elucidate the metabolic pathways involved in freezing stress, 20 pathway items with the most significant enrichment were selected from the five freezing treatments of *A. pseudosieboldianum* to be displayed in the diagram (Figure 5). Among these top pathways, carbohydrate metabolism, galactose metabolism, fructose, and mannose metabolism are associated with many ascending/descending genes. In addition, in the comparison of the five groups of experimental samples, *A. pseudosieboldianum* was enriched in some pathways under all freezing stress, but there were some special pathways in each period, such as the synthesis and degradation of ketone bodies, which were significantly enriched only at −10 °C. The MAPK signaling pathway was significantly enriched under all freezing stress, ABC transport was enriched at −10 °C, −20 °C, and −40 °C, and Limonene and pinene degradation was enriched only at −40 °C. In short, many genes from different metabolic pathways are involved in the regulation of *A. pseudosieboldianum* freezing stress, and further study on the differential expression patterns of these pathways is of great significance in revealing the cold resistance of *A. pseudosieboldianum*.

### 2.5. Identification and Analysis of Genes Associated with the MAPK Pathway

The mitogen-activated protein kinases (MAPK) signaling pathway is one of the important pathways in the eukaryotic signaling network and is a key signaling pathway for cell proliferation, differentiation, apoptosis, and stress responses under normal and pathological conditions (Figure 6). As shown in Figure 6, genes on MAPK produced upregulated or downregulated expression under freezing stress at different temperatures. Among them, upregulated genes produced up to 1.69-fold expression and downregulated genes produced up to 3.06-fold expression (Appendix A).

### 2.6. Identification and Analysis of Genes Related to Sugar Metabolism

Sugars play a crucial role in the cold resistance of plants. In plant response to freezing stress, the accumulation of soluble sugars is usually detected. Soluble sugars can act as nutrients, osmoprotectants, and cryoprotectants to protect plant cells from damage caused by freezing stress. As shown in Figure 7, genes on the sugar metabolism pathway produced significant expression under freezing stress conditions at different temperatures (Appendix A).

### 2.7. Key Transcription Factors Associated with Freezing Stress

In this study, we first determined all transcription factors and further analyzed the expression of key transcription factors (MYB, WRKY, AP2/ERF-ERF) (Figure 8) that are involved in freezing stress. The FPKM values of these transcription factors are visible in the Appendix A (Appendix A). In total, we found 115 MYB family members, 108 AP2/ERF-ERF family members, and 66 WRKY family members in all libraries (Figure 9).

## 3. Discussion

With the gradual improvement of living standards, the construction of an ecological environment is becoming more and more important. Foliage plants can not only be used as timber forests but also play an important role in urban ecological construction. Due to the influence of the relatively arid climate and low temperature in northern China, the distribution of colorful foliage plants in northern China is relatively small, and *A. pseudosieboldianum*, as an excellent colorful page plant in the north, is important to study in relation to its freezing stress. Meanwhile, due to the frequent occurrence of extreme weather in recent years, climate affects the development of many plants and even leads to death. The study of *A. pseudosieboldianum*’s response to abiotic stress also becomes very important. Multiple responses to freezing stress have been reported for many plant species, reflected in transcriptional or metabolic changes [37]. Recently, significant progress has been made in understanding the perception and transduction of cold signals in plants; however, we are still far from fully understanding the molecular mechanisms of cold signal perception and transduction in plants [38,39,40]. In our study, the genes responsible for resistance to freezing stress were identified by comparing the transcriptome profiles of *A. pseudosieboldianum* under freezing stress at different temperatures. We performed a more comprehensive analysis of the changes in important metabolic pathways and transcription factors in *A. pseudosieboldianum*. In addition, our study provides reference value for other woody plants and provides theoretical support and facilities for future development of breeding to improve the breeding of cold-tolerant species of *A. pseudosieboldianum*. In this study, the freezing stress of *A. pseudosieboldianum* was studied from three aspects: hormone, enzyme activity and transcriptome (Figure 10).

### 3.1. Identification of DEGs of A. pseudosieboldianum under Freezing Stress at Different Temperatures

Under freezing stress at different temperatures, 2505, 6021, 5125, and 3191 DEGs were identified, respectively, by comparison with the control (4 °C). Most of the important genes produced downregulation compared to the control. This result was consistent with the regulation of freezing stress in potatoes studied by Kou et al. [41]; however, the study of Niu et al. [42] on *Prunus persica* cold-tolerance resulted in a greater number of upregulated genes than downregulated genes for different levels of freezing stress, compared with the control group.

In addition, the GO classification of *A. pseudosieboldianum* genes under different temperature treatments was compared, and the functional categories were enriched, mainly in “metabolic processes”, “cellular processes”, and “response to stimuli”, and the KEGG pathway analysis showed that the main enriched pathways included phytohormone signaling, metabolic pathways (Appendix A), and ribosomal and secondary metabolic pathways. The comprehensive analysis of data from *A. pseudosieboldianum* under different temperature treatments identified more important DEGs involved in freezing stress and contributed to further understanding of *A. pseudosieboldianum*. In key gene families, many genes were significantly upregulated at CK (4 °C), −20 °C, −30 °C. However, genes were not significantly upregulated at −40 °C, which may be due to the fact that gene families such as MYB have begun to regulate under low temperature stress. The regulation is still obvious at −20 °C and −30 °C, and it is no longer regulated at −40 °C, which may be related to the extreme temperature of gene regulation of *A. pseudosieboldianum* in response to freezing stress. Meanwhile, the upregulation and downregulation changes of these three key gene families may be related to abiotic stress in *A. pseudosieboldianum*.

### 3.2. Genes Involved in the Sugar Metabolism Pathway under Freezing Stress

Sugars play a crucial role in the cold tolerance of plants. In plant response to cold freezing stress, the accumulation of soluble sugars is usually detected. Soluble sugars can act as nutrients, osmoprotectants, and cryoprotectants to protect plant cells from cold freezing stress damage. Sugars can also act as signaling molecules, similar to hormone, involved in plant growth, development, and various stress responses [43,44,45]. Therefore, several genes related to sugar metabolism were explored in this study. Sucrose, cottonseed sugar, glucose, and fructose accumulate under cold freezing stress and are involved in plant cold resistance [43,46]. Sucrose synthase and sucrose phosphate synthase are two important enzymes involved in sucrose metabolism and biosynthesis, respectively [47]. Here, the induction of 10 genes encoding sucrose synthases may point to the degradation of sucrose to UDP-glucose and fructose. This result shows the induction of sucrose synthase gene expression under freezing stress. Sucrose content and cold tolerance are positively correlated in woody plants [44,48]. Meanwhile, as shown in Figure 6, a total of 524 coding genes were involved in sugar metabolism, including 2.4.1.123 (inositol 3-alpha-galactosyltransferase), 2.4.1.67 (stachyose synthetase), 3.2.1.26 (Alkaline/neutral invertase), and 2.4.1.82 (raffinose synthase) in *A. pseudosieboldianum* at −20 °C and −30 °C. This demonstrated that sugar metabolism in *A. pseudosieboldianum* was about to be affected by freezing stress. This may be because *A. pseudosieboldianum* needs sugar to provide energy in the process of resisting freezing stress, so under the conditions of −20 °C and −30 °C, the genes are significantly upregulated, but under the freezing environment of −40 °C, the gene for *A. pseudosieboldianum* was no longer upregulated, probably because the enzyme had been inactivated in the plant.

### 3.3. Related Genes on MAPK Signaling Pathway under Freezing Stress

When growth and developmental or environmental signals occur and are delivered to effector cells, receptors located on the cell membrane, as well as intracellular signaling components, are required to deliver the signals to target sites in the cytoplasm or nucleus to elicit a cellular response. MAPK is an important component of the signaling pathway in eukaryotes, consisting of three cascades: MEKK (MAPKKK, MAP3K), MEK (MPKK, MAP2K), and MPK (MAPK), which are activated by phosphorylation, then, in turn, phosphorylate the downstream components to pass the signal on. When cells receive external signals, the receptors on the cell membrane are activated first, then the MEKK→MEK→MPK cascade is activated sequentially, and finally, MPK activates downstream targets, such as enzymes and transcription factors, causing a cellular response (Figure 1 and Figure 2). MAPK is involved in a wide range of physiological processes in plants, including growth and development and plant response to biotic and abiotic stresses [49,50].

From Figure 5, we can see that *A. pseudosieboldianum* showed significant gene expression in pathogen defense (PR1) and stress adaptation (MAP3K18) under the effect of freezing stress. The highest gene expression was observed insStress adaptation (MAP3K18) and pathogen defense (PR1) under −40 °C treatment. It can be seen from the above that the MAPK pathway still plays a significant role in responding to extremely low temperature environments, and it is upregulated at −20 °C, −30 °C and −40 °C.

### 3.4. Transcriptional Regulation-Related Genes under Freezing Stress

Transcription factors play important functions in abiotic stresses in plants (Appendix A and Appendix A). It has been shown that in Arabidopsis, most cold-regulated transcription factor genes are impaired in the plant response to cold stress [51,52,53]. It has also been shown that many transcription factors produce important roles in plant cold stress [54,55]. For example, plants induce the expression of cold-regulated genes (*COR*) to withstand freezing stress. *COR* genes in Arabidopsis include *COR,* LTI low-temperature induced (LTI) genes, responsive to desiccation (RD) genes, and early dehydration-inducible (ERD) genes, which encode key enzymes for the synthesis of osmoregulatory substances that increase tolerance by CBFs/DREB1 recognizing and binding the CRT/DRE cis-acting element containing the conserved sequence of CCGAC on the *COR* gene promoter to regulate the expression of *COR* genes. When plants are exposed to freezing stress, *CBFs* genes are rapidly induced to be expressed, and subsequently, *COR* gene expression is activated. For example, the AP2/ERF factors RAP2.1 and RAP2.6 and the C2H2-type zinc finger STZ/ZAT10 belong to the *CBF* regulators [56,57]. Many MYB-type transcription factors, such as *AtMYB85* [58] and *AtMYB88* [59], increase chilling tolerance in apple through CBF-dependent and CBF-independent pathways. In this study, we analyzed the expression patterns of the MYB, WRKY, and AP2/ERF-ERF gene families and found that different members play different roles under different levels of freezing stress. For example, in the WRKY family, different members of the family were involved in the response to low temperature.

Among these gene families, *Apse001G0028200* and *Apse002G0128400* showed significant changes in freezing stress treatment at −30 °C; *Apse011G0120800* showed significant changes in freezing stress treatment at −30 °C and −40 °C. These differences suggest that *A. pseudosieboldianum* has significant changes in transcription factors under different levels of freezing stress, thus verifying that these gene families may have regulatory mechanisms in the freezing stress of *A. pseudosieboldianum*. From the experimental results, it can be concluded that these differential genes may be related to the response of *A. pseudosieboldianum* to freezing stress. Among them, the *Apse011G0120800* gene may be related to the response to extremely low temperature. Many genes are not upregulated at −40 °C, which may be caused by the extreme low temperature environment, in which the freezing resistance gene of *A. pseudosieboldianum* could no longer be expressed.

### 3.5. Changes in Physiological Indicators of A. pseudosieboldianum under Different Temperatures of Freezing Stress

The mechanisms that respond to low temperature stress in plants are complex and diverse. Once the low temperature signal is received, defense mechanisms are turned on, including physical structural adaptation (e.g., changes in membrane conformation), in-tercellular regulation of osmotic substances (e.g., Ca^2+^, soluble sugars, proline, betaine) and scavenging of reactive oxygen species by protective enzyme systems (POD, SOD, others) [60]. In this study, the physiological responses of *A. pseudosieboldianum* under different levels of freezing stress were observed by measuring the activities of five enzymes, SOD, POD, CAT, APX and GSH-Px, and the contents of five hormones and metabolites, ABA, MDA, H_2_O_2_, proline and soluble sugar, at different temperatures of freezing stress. The physiological responses of *A. pseudosieboldianum* under different temperatures of freezing stress were observed. ABA is a gibberellin and cytokinin inhibitor, which is related to leaf senescence and fruit abscission. Freezing stress can induce the accumulation of ABA, and ABA can induce the production of new proteins, which are involved in regulating the content of polyamines and soluble carbohydrates, increasing the activity of antioxidant enzymes, and enhancing the cold tolerance of plants [61,62,63]. From Figure 7, we can see that the activities of POD, CAT, and APX gradually increased until −30 °C and reached the highest at −30 °C, while SOD had the highest activity at −20 °C. The hormone contents of ABA, MDA, H_2_O_2_, proline, and soluble sugar gradually increased until −30 °C, reached the highest at −30 °C, and decreased immediately thereafter. From the graph, we speculate that in −40 °C the enzyme activity was reduced due to the reduced activity of *A. pseudosieboldianum* samples and the hormones could not continue to accumulate. As can be seen from the slope of the line graph, in response to freezing stress, the activities of POD, CAT and APX increased significantly in CK (4 °C) ~ −20 °C, and responded rapidly to freezing stress, the activity of SOD increased significantly at CK (4 °C) ~ −10 °C, and responded rapidly to freezing stress.; the activity decreased significantly at −40 °C, possibly due to the inactivation of the enzyme due to extreme low temperature. In endogenous hormones, ABA, MDA, H_2_O_2_, proline, and soluble sugar accumulates rapidly in plants at CK (4 °C)~−30 °C. In this study, at −40 °C, it dropped significantly again, which may be because the genes that control endogenous hormones in *A. pseudosieboldianum* are no longer upregulated at −40 °C.

## 4. Materials and Methods

### 4.1. Plant Materials

Four-year-old *A. pseudosieboldianum* seedlings were collected from the greenhouse of Northeast Forestry University. The collection was made on 19 November 2021, at a temperature ranging from 8 °C to 4 °C. Forty pots of current-year branches of four-year-old *A. pseudosieboldianum* seedlings of approximately the same growth were taken. The collected branches were left at 4 °C and 0 °C for 10 h and used as a control (APCK), while the other branches were treated and kept at freezing temperatures for 10 h: −10 °C (APA), −20 °C (APB), −30 °C (APC), and −40 °C (APD). In each treatment, 20 branches were divided into four groups, three as transcriptional samples and one as a physiological and biochemical sample. The xylem of the branches was then collected and snap-frozen in liquid nitrogen.

### 4.2. Determination of Physiological Indicators

In this study, the hormone contents and enzyme activities of five enzymes were determined in five groups of *A. pseudosieboldianum* under freezing stress at different temperatures. ABA levels were determined by double antibody sandwich assay; MDA, soluble sugar, proline, and H_2_O_2_ were determined by double antibody one-step sandwich enzyme-linked immunosorbent assay. COD, SOD, CAT, APX, and GSH-Px were determined by double antibody one-step sandwich enzyme-linked immunosorbent assay.

### 4.3. RNA Library Construction and Sequencing

The starting RNA for library construction was total RNA, ≥1 μg. The library was built using Illumina’s NEBNext^®^ UltraTM RNA Library Prep Kit. mRNA with polyA tails was enriched by Oligo(dT) beads, and the resulting mRNA was randomly interrupted with divalent cations in NEB fragmentation buffer. The fragmented mRNA was used as a template and random oligonucleotides were used as primers to synthesize the first strand of cDNA in the MMuLV reverse transcriptase system, followed by degradation of the RNA strand with RNaseH and synthesis of the second strand of cDNA with dNTPs in the DNA polymerase I system. The purified double-stranded cDNA was end-repaired, A-tailed, and sequenced, and the cDNA of approximately 200bp was screened with AMPure XP beads, PCR amplified, and the PCR products were purified again with AMPure XP beads to obtain the final library. After the library was constructed, it was initially quantified using a Qubit2.0 fluorometer, diluted to 1.5 ng/μL, and then the insert size of the library was measured using an Agilent 2100 bioanalyzer. After the insert size met the expectation, qRT-PCR was performed to accurately quantify the effective library concentration (i.e., the effective library concentration above 2 nM) to ensure the quality of the library.

After passing the library check, the different libraries were pooled by effective concentration and the target downstream data volume, then sequenced by Illumina, and 150 bp paired-end reads were generated. The basic principle of sequencing is sequencing by synthesis. Four fluorescently labeled dNTPs, DNA polymerase, and splice primers were added to the sequencing flow cell for amplification. When each sequencing cluster extended the complementary strand, each fluorescently labeled dNTP was added to release the corresponding fluorescence, and the sequencer captured the fluorescence signal and converted the light signal into sequencing peaks through computer software to obtain sequence information about the fragment to be sequenced.

### 4.4. Comparison of Transcriptome Data with the Reference Genome

Sequenced fragments are randomly interrupted by mRNA. To determine which genes these fragments are transcribed from, clean reads from quality control need to be matched to the reference genome. The clean reads are sequenced against the reference genome using HISAT2 (https://download.cncb.ac.cn/gwh/Plants/Acer_pseudosieboldianum_Apse_genome_GWHBECT00000000/GWHBECT00000000/GWHBECT0000.genome.fasta.gz, accessed on 6 October 2022) [64,65] to obtain information on the position of the reference genome or gene, as well as information on the. sequence characteristics specific to the sequenced sample. The algorithm of HISAT2 [65,66] is divided into three main parts: (1) whole segment alignment of sequenced sequences to single exons of the genome; (2) segment alignment of sequenced sequences to two exons of the genome; and (3) segment alignment of sequenced sequences to three or more exons of the genome.

### 4.5. Differential Gene Screening

DESeq2 [67,68] was used to analyze genes expressed differentially among the sample groups, and the differentially expressed genes between the two biological conditions were obtained. After inputting the unstandardized reads counting data of genes, the multiple hypothesis test and correction of the hypothesis test probability (P value) was carried out by the Benjamini–Hochberg method, and the error detection rate (false discovery rate, FDR) was obtained. The screening condition of differential genes was |log2Fold Change| ≥ 1, and FDR < 0.05.

### 4.6. RT-qPCR Validation

RNA-seq data were validated by RT-qPCR analysis (Appendix A). Amplification was performed using a 7500 Fast Real-Time PCR System and SYBRP remix EXTaq kit (Takara, Kyoto, Japan). The reaction protocol was as follows: 95 °C for 30 s, 45 cycles, 95 °C for 5 s, 59 °C for 15 s, 72 °C for 20 s, and 72 °C for 7 min. Appendix A lists all the specific primers used. Three technical replicates were used for each sample (Appendix A). ACTIN (accession number: AY261523.1) was used as a quantitative control to determine the relative expression values according to the 2^−ΔΔct^ algorithm [69].

## 5. Conclusions

In this study, transcriptome sequencing was first used to analyze *A. pseudosieboldianum* under different temperature freezing stresses at different temperatures. We considered the important genes involved in signal transduction, transcription, and regulation under freezing stress at different temperatures.

There was differential expression in *A. pseudosieboldianum*. Some different members of the MYB, WRKY, and AP2/ERF-ERF gene families were significantly upregulated/downregulated under different-temperature freezing stress. In addition, some DEGs involved in glucose metabolism, such as *Apse010G0038900* and *Apse001G0122200*, were significantly upregulated during freezing at −30 °C, which further emphasized the importance of glucose metabolism in plant freezing tolerance.

As an important part of the signal transduction pathway in eukaryotes, MAPK also played an important role in freezing stress. Among the genes involved in MAPK signaling pathway, the genes treated at −40 °C were significantly upregulated in MAP3K18 and PR1.

In the physiological indicators of hormone and enzyme activity, *A. pseudosieboldianum* with the decrease of treatment temperature, enzyme activity, and hormone accumulation increased gradually. Under the freezing stress of −30 °C, the activities of POD, CAT, and APX were the highest, and the accumulation of ABA, MDA, H_2_O_2_, proline, and soluble sugar was the highest. When the temperature was lower than −30 °C, the enzyme activity and hormone level decreased gradually, which was speculated to be caused by the decrease in the main body activity of *A. pseudosieboldianum* sample.

The results of this study can explain the molecular mechanism and play an important role in the further study of the cold tolerance of plants. The important DEGs found in this study may be helpful in further studying the freezing tolerance mechanism of *A. pseudosieboldianum* and other woody plants.

## Figures and Tables

**Figure 1 ijms-23-14676-f001:**
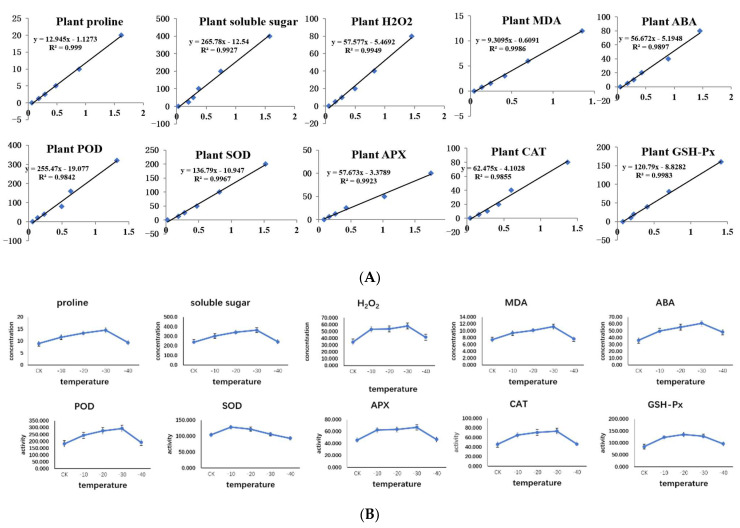
The expression of physiological indicators of *A. pseudosieboldianum* under freezing stress at different temperatures. (**A**) is the curve analysis of different physiological indicators to determine the accuracy of the test data; (**B**) is the activity and content expression of physiological indicators under freezing stress at different temperatures. These include the enzymatic activities of SOD, POD, CAT, APX and GSH-Px and the accumulation of ABA, MDA, H_2_O_2_, proline, and soluble sugar under freezing stress at different temperatures.

**Figure 2 ijms-23-14676-f002:**
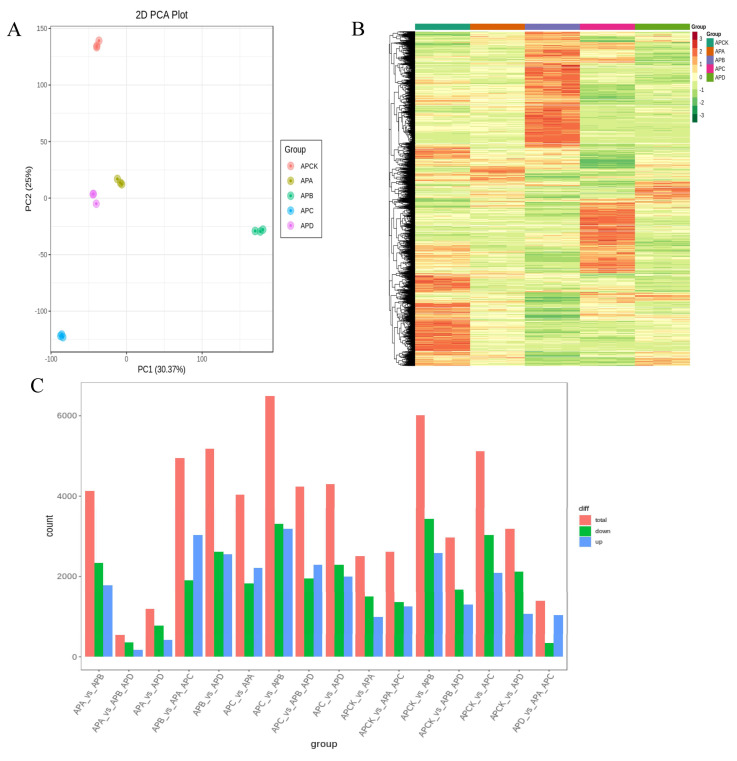
The number of DEGs caused by freezing of *A. pseudosieboldianum* at different temperatures. Map An is principal component analysis (PCA), which compresses the original data into five principal components to describe the characteristics of the original data set (**A**). PC1 represents the most obvious features in the multidimensional data matrix, PC2 represents the most significant features in the data matrix except PC1, (**B**) shows the clustering analysis of gene expression in five periods, and (**C**) shows the quantitative expression of differential genes in different periods. APCK, APA, APB, APC, and APD represent *A. pseudosieboldianum*, 4 °C, −10 °C, −20 °C, −30 °C and −40 °C, respectively.

**Figure 3 ijms-23-14676-f003:**
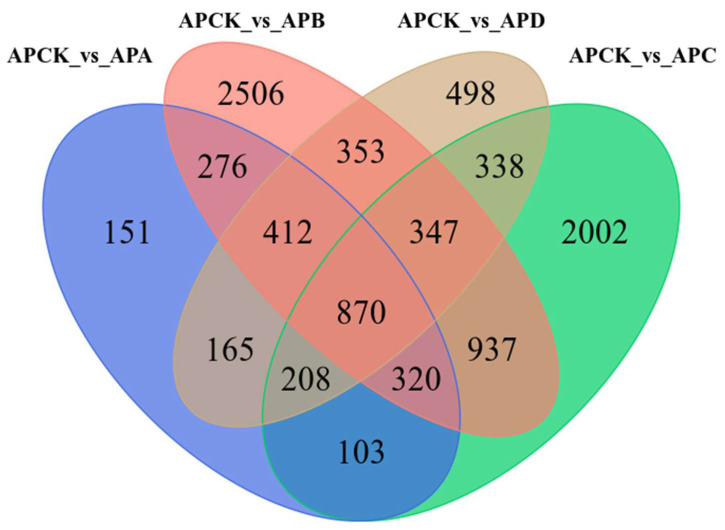
Wayne diagram of *A. pseudosieboldianum* and DEGs of CK (4 °C) under different temperature freezing stress. They provide a comparison between −10 °C treatment and 4 °C treatment, −20 °C treatment and 4 °C treatment, −30 °C treatment and 4 °C treatment, and −40 °C treatment and 4°C treatment.

**Figure 4 ijms-23-14676-f004:**
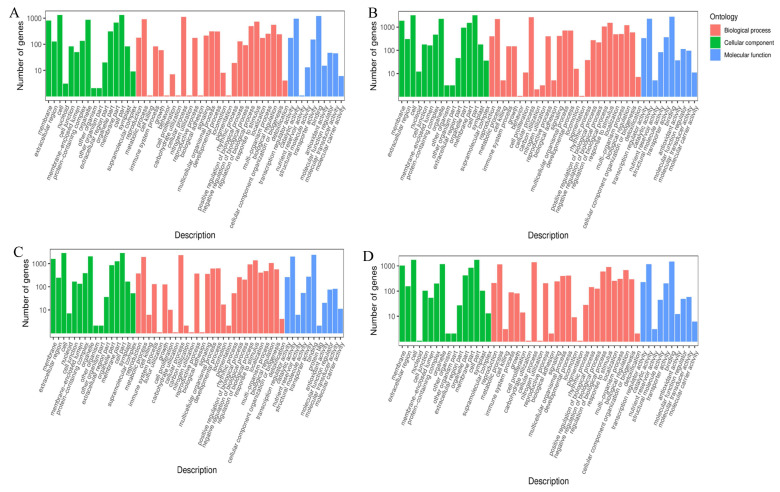
GO analysis of DEGs of *A. pseudosieboldianum* under freezing stress at different temperatures. The Y and X axes correspond to GO terminology and the number of DEGs. The (**A**) figure shows the comparison of −10 °C treatment and 4 °C treatment; the (**B**) diagram shows the comparison of −20 °C treatment and 4 °C treatment; the (**C**) diagram shows the comparison of −30 °C treatment and 4 °C treatment; and the (**D**) diagram shows the comparison of −40 °C treatment and 4 °C treatment.

**Figure 5 ijms-23-14676-f005:**
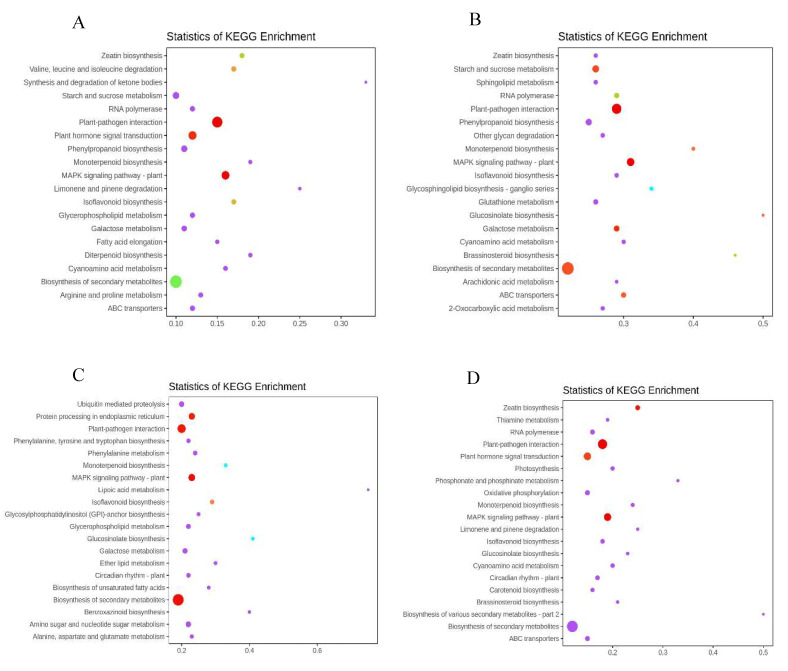
KEGG enrichment analysis of DEGs in *A. pseudosieboldianum* under freezing stress at different temperatures. KEGG pathway classification of DEGs of *A. pseudosieboldianum* under low-temperature stress at different temperatures and CK group control, respectively. The *y*-axis corresponds to the KEGG pathway, the *x*-axis indicates the number of DEGs enriched in a pathway between the enrichment rate. The color of the dot represents the q-value and the size of the dot represents the number of DEGs mapped to the reference pathway. A larger rich factor indicates a greater enrichment. A smaller q-value indicates a more significant enrichment. (**A**) is the KEGG enrichment analysis of DEGs between APCK (4 °C) and APA (−10 °C), (**B**) is the KEGG enrichment analysis of DEGs between APCK (4 °C) and APB (−20 °C), (**C**) is the KEGG enrichment analysis of DEGs between APCK (4 °C) and APC (−30 °C), (**D**) is the KEGG enrichment analysis of DEGs between APCK (4 °C) and APD (−40 °C).

**Figure 6 ijms-23-14676-f006:**
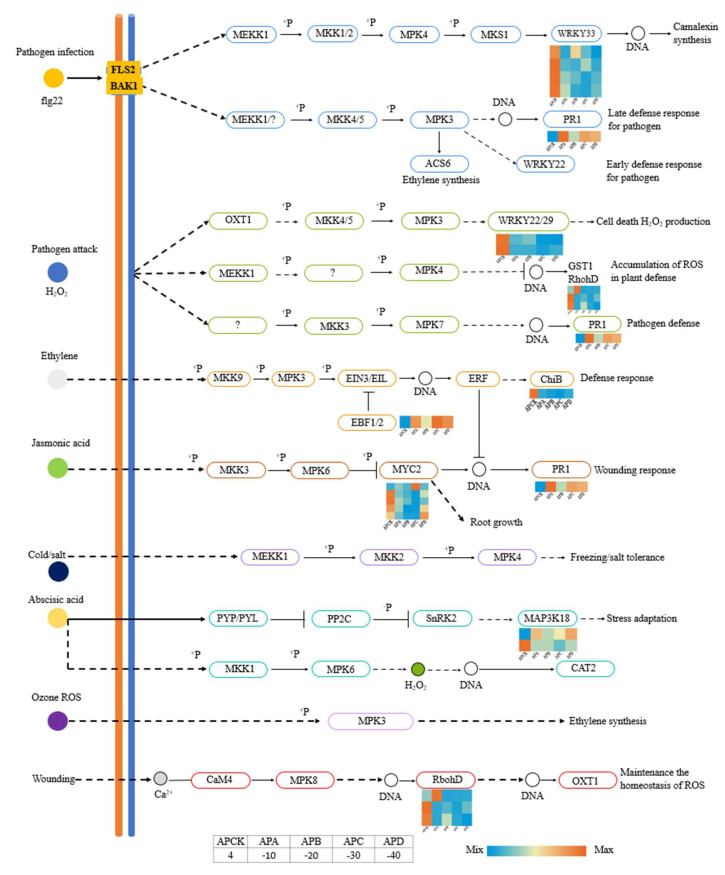
Expression of DEGs on MAPK pathway in *A. pseudosieboldianum* under freezing stress at different temperatures. The heatmap expression shows the differential gene expression of *A. pseudosieboldianum* under freezing stress at different temperatures compared to CK (4 °C).

**Figure 7 ijms-23-14676-f007:**
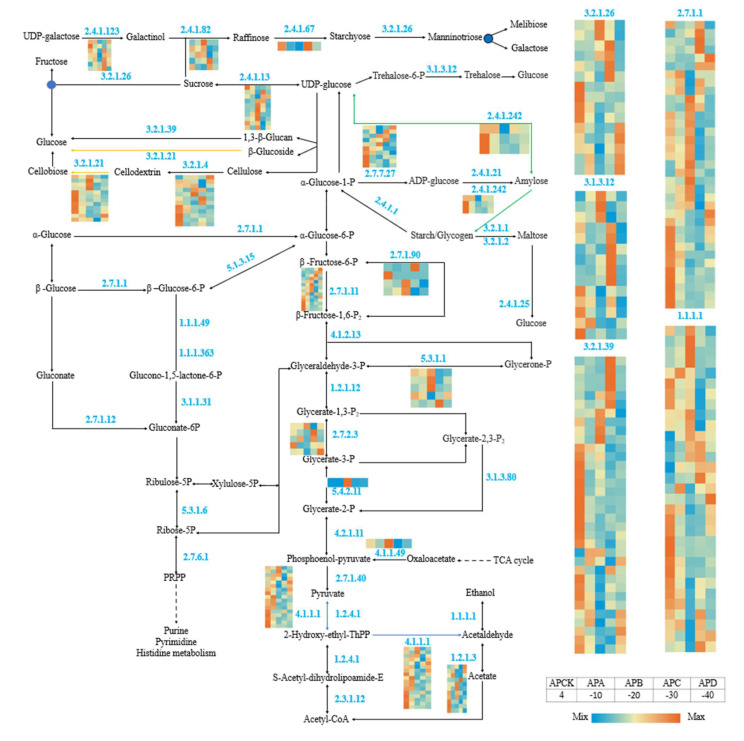
Expression of DEGs in the gluconeogenic pathway of *A. pseudosieboldianum* under freezing stress at different temperatures. The heat map shows the differential gene expression of *A. pseudosieboldianum* under freezing stress at different temperatures compared to CK (4 °C).

**Figure 8 ijms-23-14676-f008:**
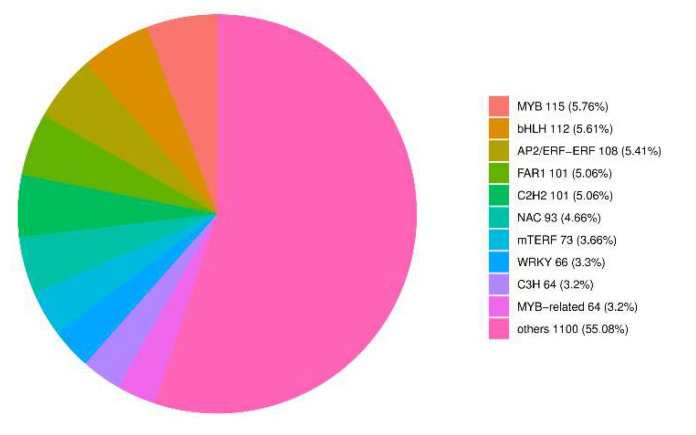
Transfer factors of *A. pseudosieboldianum* under freezing stress at different temperatures.

**Figure 9 ijms-23-14676-f009:**
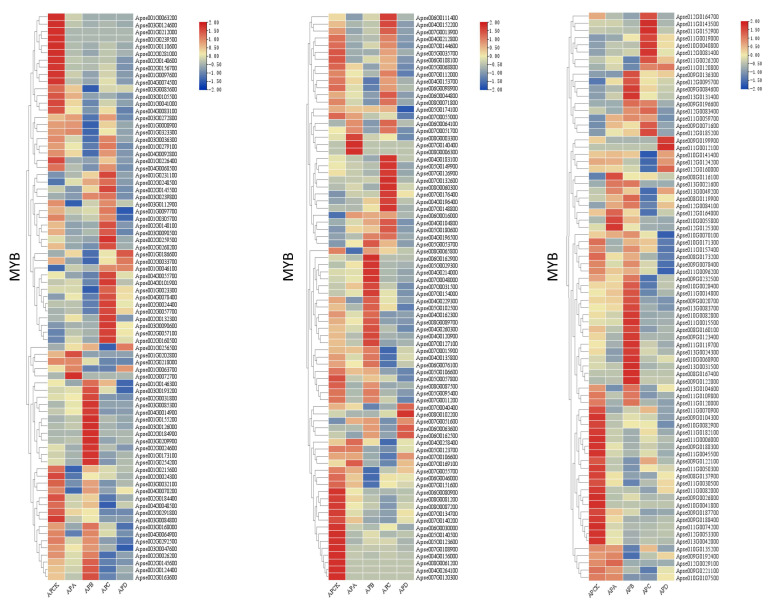
Expression of MYB, AP2/ERF, and WRKY genes. APCK: *A. pseudosieboldianum* treated at 4 °C; APA: *A. pseudosieboldianum* treated at −10 °C; APB: *A. pseudosieboldianum* treated at −20 °C; APC: *A. pseudosieboldianum* treated at −30 °C; APD: *A. pseudosieboldianum* treated at −40 °C.

**Figure 10 ijms-23-14676-f010:**
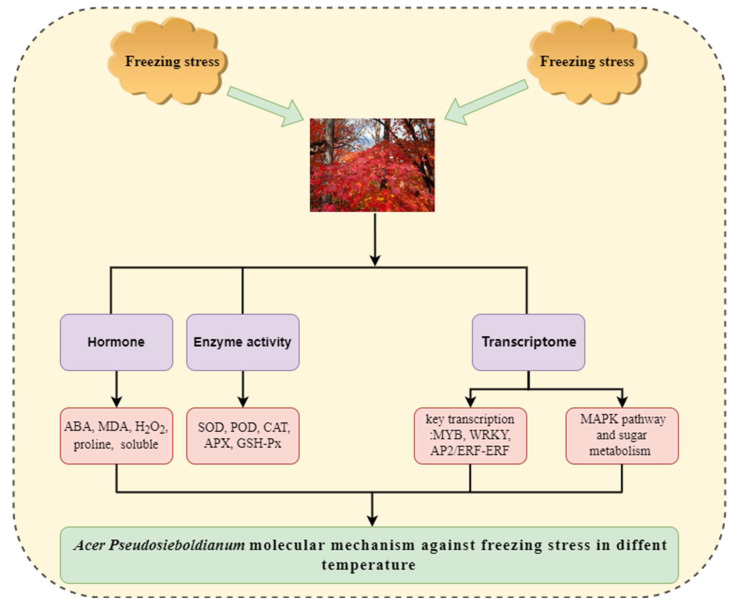
*A. pseudosieboldianum* freezing stress summary figure.

**Table 1 ijms-23-14676-t001:** Summary of the RNA-seq data collected from *A. pseudosieboldianum*.

Sample	Raw Reads	Clean Reads	Clean Base (G)	Q20 (%)	Q30 (%)	GC Content (%)	Reads Mapped	Unique Mapped
APCK-1	67,112,578	62,694,296	9.4	98.05	94.31	44.34	54,661,894 (87.19%)	51,783,721 (82.60%)
APCK-2	72,192,138	67,23,1322	10.08	98.11	94.53	44.42	58,666,971 (87.26%)	55,645,093 (82.77%)
APCK-3	77,334,844	72,727,740	10.91	98.18	94.66	44.52	63,647,230 (87.51%)	60,429,351 (83.09%)
APA-1	69,972,072	64,699,338	9.7	98.16	94.61	44.58	57,852,215 (89.42%)	54,562,832 (84.33%)
APA-2	68,091,786	64,238,838	9.64	98.13	94.52	44.61	57,515,124 (89.53%)	54,348,737 (84.60%)
APA-3	66,558,226	62,252,698	9.34	98.19	94.72	44.65	55,850,429 (89.72%)	52,916,333 (85.00%)
APB-1	65,941,198	63,454,732	9.52	97.28	92.85	44.04	56,615,003 (89.22%)	53,874,843 (84.90%)
APB-2	52,211,874	50,641,818	7.60	97.27	92.80	43.83	45,193,337 (89.24%)	43,078,994 (85.07%)
APB-3	57,659,774	56,777,300	8.52	97.04	92.15	43.77	50,733,924 (89.36%)	48,444,786 (85.32%)
APC-1	61,411,604	54,530,820	8.18	98.21	94.78	44.48	49,423,885 (90.63%)	46,837,461 (85.89%)
APC-2	64,932,060	61,789,950	9.27	98.18	94.68	44.07	56,368,800 (91.23%)	53,583,580 (86.72%)
APC-3	67,066,254	63,155,790	9.47	98.15	94.59	44.15	57,620,465 (91.24%)	54,696,618 (86.61%)
APD-1	59,016,432	56,378,222	8.46	97.95	94.17	44.93	50,014,664 (88.71%)	47,167,630 (83.66%)
APD-2	49,445,182	46,892,166	7.03	98.21	94.77	44.87	41,665,473 (88.85%)	39,416,305 (84.06%)
APD-3	60,492,000	56,359,986	8.45	98.21	94.77	44.85	50,029,208 (88.77%)	47,315,447 (83.95%)

APCK was treated at 4 °C of *A. pseudosieboldianum*, APA at −10 °C, APB at −20 °C, APC at −30 °C, and APD at −40 °C. Reads mapped is the number of reads compared with the reference genome, and unique mapped is the only number of reads that is aligned with the reference genome.

## Data Availability

The *A. pseudosieboldianum* raw genome sequencing data are available from the NCBI under project ID PRJCA006356. *A. pseudosieboldianum A. pseudosieboldianum* Transcriptome or Gene expression in freezing stress from the NCBI under project ID PRJNA894936.

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
