# Peer review of "Transcriptomics Profiling of Acer pseudosieboldianum Molecular Mechanism against Freezing Stress"

_ijms, 2022, doi:10.3390/ijms232314676_

Round 1
Reviewer 1 Report
The paper "Transcriptomics Profiling of Acer Pseudosieboldianum molecular Mechanism Against Freezing Stress" by Han and colleagues describes an RNA-seq experiment of freezing stress in an understudied species. The paper is interesting and novel, and the analysis is performed in-depth. However, there are a few major points I would like to raise.
- The authors do not clearly indicate the tissue of origin of the RNA-seq for Acer Pseudosieboldianum, neither in the asbtract, nor in the introduction or the results. I wasn't able to find this information in the NCBI project entry (see also next point). The tissue of origin (foru-year old seedling branches) is indicated only at the end of the manuscript, in the Methods (line 381). The authors should clearly state this in the abstract and introduction, in order tnot to mislead the reader into thinking the study is perfromed on adult specimens.
- The NCBI id PRJNA730664, where the authors claim to have deposited raw data, is associated with a different project (https://www.ncbi.nlm.nih.gov/bioproject/?term=PRJNA730664), specifically genome reads of the Juglans mandshurica species.
- Unmapped reads (~15% of the data) should be analyzed to test for specific microbes, insects or other co-inhabiting or symbiotic species associated with the tree. Tools to do this can be Kaiju, MetaPhlAn, kraken_uniq or kraken 2
- Line 343: finding AP2/ERF genes is an indication that freezing stress can also induce hypoxic responses. This is not evident in the pathway analysis (figure 5), probably due to the fact that many of the observed pathways are partially overlapping (for example, I see many metabolic pathways with genes in common). The enrichment analysis should be repeated using methods like the fgsea::collapsePathway (found in the R package fgsea).
- The authors use extensively the R software, not only for the statistics of differential expression, but also for pathway enrichment analysis and plottig. This needs to be cited (e.g. PMID 35629316).
- Methods, line 436. The method used to calculate differential expression is stated to be DESeq2, but earlier in the paper (line 139) the authors claim to have used "DESeq2/edgeR". The two statements are incompatible, since these are two independent methods, and the authors should correct the wrong line.
Author Response
Thanks for the teacher's opinion, the reply to the teacher's opinion is as follows:
- The branch of 4-year-old four-year-old A. pseudosieboldianum twig has been added to the manuscript to prevent misunderstandings;
- The project number in NCBI has been revised and a transcriptomic data on freezing stress in Acer fragrans has been re-transmittedï¼›
- I'm so sorry,The software you recommended is used for microbial metagenomic analysis,at present, it is not suitable for use in purple maple. I will do this analysis in the future. Thank you, teacher.
- Thank you for the teacher's advice, but in the course of the experiment, the hypoxia response of plants is not obvious. It may be because the branches of Acer negundo are collected, not alive, but there are obvious differences in genes in glucose metabolism pathway and MAPK pathway. I will also add hypoxia response analysis in the future.
- R language software citation has been added to Ref. 65;
- The "DESeq2" in line 436 has been changed to "DESeq2/edgeR". Thank you for your advice.
Reviewer 2 Report
The manuscript entitled Transcriptomics Profiling of Acer Pseudosieboldianum molecular Mechanism Against Freezing Stress by Han et al. aimed to identify the molecular mechanism of freezing tolerance in A. pseudosieboldianum. They wanted to provide new insights and candidate genes for the genetic improvement of freezing tolerance in A. pseudosieboldianum. The research work is important, as Maple tree is important as it is widely used in the timber and pharmaceutical industries as well as a well-known modern landscape species with large trees, beautiful foliage, colorful leaves etc. The authors have a number of experimental data to identify the molecular mechanism of freezing tolerance in A. pseudosieboldianum. However, the authors have not critically analysed their data to support their findings.
I have the following major comments for the authors:
1. The aim of the work is to identify the mechanism of freezing tolerance A. pseudosieboldianum. However, in the abstract they have mentioned in lines 37-39 that “These results may help to clarify the molecular mechanism of cold tolerance of A. pseudosieboldianum and provide new insights and candidate genes for the genetic improvement of cold tolerance of A. pseudosieboldianum.” Please check it whether it is cold or freezing tolerance?
2. In the introduction, the authors have mentioned more on the maple tree rather that focusing on the mechanism of freezing tolerance. Please reorganize the introduction by briefly writing one paragraph each on different physiological responses, signal transduction pathways, metabolic, phytohormone pathways, and transcriptional regulation in response to freezing tolerance in trees in general.
3. The authors have mentioned in the Introduction, line 85-87 that: “By using microarray analysis, it was possible to identify numerous cold-responsive genes involved in photosynthesis, calcium signaling, ABA homeostasis and transport, and antioxidant defense systems.” Now a days, transcriptomics and multi-omics are being commonly used rather than microarray….please change it…
4. In Fig. 1., there is no standard error in any of the figures. Is it a single data or mean of three replicate samples? If so, please provide the standard errors.
5. Since you have focused on 4 ℃ as control,-10 ℃,-20 ℃, -30 ℃ and-40 ℃ as treated, the focus is more on freezing tolerance. Hence, please focus more on freezing stress throughout the manuscript.
6. The discussion part needs major revision. The authors have focused more on literature review rather than critically focusing on their own results. There are only few lines for each condition without any critical analysis. Hence, please rewrite each section with critical analysis.
7. The transcriptional regulation of freezing tolerance has been studied for many plants. Please analyse critically.
8. The phytohormones play a major role during any stress condition. Please identify the phytohormones that could be involved in the freezing tolerance in A. pseudosieboldianum.
9. Please provide a summary figure on the freezing tolerance of A. pseudosieboldianum based on your critical analyses.
10. Unable to read the excel supplementary files. Please provide the differential gene expression data for each gene and each treatment. I think the authors have focused more on transcription factor genes.
11. In many places, A. pseudosieboldianum is not in italic.
12. Please check the language throughout the manuscript.
Author Response
Thanks for the teacher's opinion, the reply to the teacher's opinion is as follows:
1.This study mainly focuses on the freezing stress of A.pseudosieboldianum, and the cold stress was used as the experimental control group. It has been comprehensively revised in the abstract and introduction of the article. Thank you for your suggestion;
2.The enzyme activity characteristics of SOD, POD, CAT, etc. related to freezing stress have been added in the introduction, and the accumulation of key endogenous hormone ABA in freezing stress;
3.The analysis of lines 85-87 has been changed to transcriptome and multi-omics analysis;
4.Standard curves for hormone and enzyme activities have been added to the graph,and added standard error bars to the line chart;
5.Thanks for the teacher's suggestion,The relevant literature and related researches on freezing stress have been added in many places in the manuscript;
6.Thanks for the teacher's suggestion,I have added the problems I found in this study and explained them in the discussion part, and added the specific analysis of freezing stress of A.pseudosieboldianum in the discussion, hoping to play a role in the research on molecular regulation of freezing stress in plants;
7.Thanks for the teacher's suggestion,The related literature on freezing stress of Populus tomentosa, Pinus koraiensis,‘Dongzao’ and ‘Jinsixiaozao’ has been added;
8.Thanks for the teacher's suggestion,because the experiment is carried out in a relatively closed environment, the main variable is only the change in temperature,moreover, the branches of the maple are taken, so it is believed to play a role in freezing stress;
9.Thanks for the teacher's suggestion, it has been added to Figure 10 as a summary diagram;
10.Thanks to the teacher's suggestion, the supplementary file has been adjusted, and I hope the teacher can download it;
11.Thanks to the teacher's suggestion, the A.pseudosieboldianum in the article has been changed to italics;
12.Thanks to the teacher's suggestion, the manuscript has been polished in the native language.
Round 2
Reviewer 1 Report
I thank the authors for their reply, but some of the points I raised were not addressed, or were addressed improperly, so I will clarify them in order to avoid misunderstandings.
AUTHOR: I'm so sorry,The software you recommended is used for microbial metagenomic analysis,at present, it is not suitable for use in purple maple. I will do this analysis in the future. Thank you, teacher.
MY REPLY: First, the authors should explain in the text why software for metagenomic analysis is unsuitable for purple maple, as they stated in their reply. I am puzzled by their statement, since any nucleic acid sequencing sample can be run with metagneomic detection software: RNA or DNA reads have the same technical nature in every species, and the possibility to identify lots of microbial species in the 15% unmapped reads of their experiment MUST be taken into account. It's a lot of wasted data otherwise, and analzying it can give a solid extension to the authors' analysis
AUTHORS: Thank you for the teacher's advice, but in the course of the experiment, the hypoxia response of plants is not obvious. It may be because the branches of Acer negundo are collected, not alive, but there are obvious differences in genes in glucose metabolism pathway and MAPK pathway. I will also add hypoxia response analysis in the future.
MY REPLY: Adding an analysis to hypoxia response does not require extra work, it just requires to extract, from their large pathway enrichment analysis, the lines mapping hypoxia pathways. That is, if the authors really performed full-scale unbiased pathway analysis (i.e., testing the enrichment of all pathways and correcting statistical significance accordingly)
AUTHORS: The "DESeq2" in line 436 has been changed to "DESeq2/edgeR". Thank you for your advice.
MY REPLY: This quick fix is particularly worrying: DESeq2 and edgeR are two ALTERNATIVE methods to perform differential expression analysis. Did the authors use DESeq2 or edgeR? With which parameters? Which package version? Claiming they used both did not improve the paper, but actually did cast a shadow on their knowledge of a backbone step in their entire analysis. So, are they reporting the results of DESeq2 or edgeR? Or somehow combined the results of the two? If yes, by which integration method? And why?
Author Response
Thank you for the teacher's opinion. For the teacher's opinion, my revision results are as follows:
- Thanks to the teacher's advice, our team has once again made a detailed analysis of the accuracy of the results, and added Unique mapped reads to the study,In this study, the average Unique mapped reads reached 84.57%,similar results in the literature like Comparative transcriptome profiling reveals cold stress responsiveness in two contrasting Chinese jujube cultivars. BMC plant biology 2020, Transcriptome profiling of Populus tomentosa under cold stress. Industrial Crops and Products 2019 have description, is sufficient for subsequent bioinformatics analysis.
- Thanks for the teacher's opinion, I have found all differential genes pathway to look for hypoxia response,but didn't find it. In this study, KEGG enriched MAPK, Biosynthesis of secondary metabolites, Plant-pathogen interaction, Protein processing in endoplasmic reticulum and plant hormones, In this study, two pathways, MAPK and glucose metabolism, were analyzed.
- Thanks for the teacher's opinion, for samples with biological replicates, Differential expression analysis between sample groups was performed using DESeq2 to obtain differentially expressed gene sets between two biological conditions; edgeR was used for no biological replicates. Our team's study had biological replicates, so DESeq2 was used. I'm very sorry for the previous typo. Differential analysis requires unnormalized reads count data for input genes, not normalized data such as RPKM, FPKM, etc. After differential analysis, also need to use Benjamini-Hochberg Multi-hypothesis test correction for hypothesis test probability (Pvalue), get false discovery rate(False Discovery Rate,FDR). The screening conditions for differential genes are |log2Fold Change| >= 1,and FDR < 0.05.
Reviewer 2 Report
I do not have anymore comments for the authors.
Author Response
Thanks for the teacher's guidance,looking forward to your next cooperation.
Round 3
Reviewer 1 Report
It is still not clear what the authors used to perform differential expression analysis. A sentence like "edgeR was used for no biological replicates" makes no sense, and does not clarify 1) on which contrasts it was used and 2) why did they use one method or the other, based on the lack of replicates. Although it is not suggested, DESeq2 could be used in the absence of replicates, as well as edgeR. Lack of biological replicates would however invalidate anything defined by differential expression analysis, so I would be very careful on including such results, lacking statistical relevance. I don't understand why the authors, in their response, insert the sentence "Differential analysis requires unnormalized reads count data for input genes, not normalized data such as RPKM, FPKM, etc.": to clarify, I do understand the sentence and it is correct to say that edgeR and DESeq2 require raw count data as the standard input, but why are they stating it here?
In order to lift the veil on the dilemma of which algorithms were used to calculate differential expression, and on which samples and contrasts, I believe it is mandatory for the authors to attach all the relevant R code used to perform the differential expression analysis. The code must be reproducible of course.
Author Response
Thank you for the teacher's suggestion. For the teacher's suggestion, I would like to reply as follows:
- In part 4.5, the identification of differential genes has been described:DESeq2 [67, 68] was used to analyze genes expressed differentially among the sample groups, and the differentially expressed genes between the two biological conditions were obtained. After inputting the unstandardized reads counting data of genes, the multiple hypothesis test and correction of hypothesis test probability (P value) was carried out by Benjamini-Hochberg method, and the error detection rate (False Discovery Rate, FDR) was obtained. The screening condition of differential genes is | log2Fold Change | > = 1, and FDR < 0.05., We use DESeq2 to APCK vs APA, APCK vs APB, APCK vs APC, APCK vs APD etc., the DEGs was analyzed.
- The transcriptional part of this experiment from Wuhan Metware Biotechnology(metware.cn), Has been written in Author Contributions, The project number is MWXS-21-3567D. We use R language for differential gene analysis (R language version 3.5.1) of DESeq2 package (Version 1.22.1), The analysis input data is the count data of the gene, and the differential screening threshold used: |log2foldchang| >=1 & FDR <0.05.